# Sacroiliac Screw Placement with Ease: CT-Guided Pelvic Fracture Osteosynthesis in the Elderly

**DOI:** 10.3390/medicina58060809

**Published:** 2022-06-15

**Authors:** Hannah Kress, Roman Klein, Tim Pohlemann, Christoph Georg Wölfl

**Affiliations:** 1Klinik für Orthopädie, Unfallchirurgie und Sporttraumatologie, Marienhaus Klinikum Hetzelstift, 67434 Neustadt an der Weinstraße, Germany; hannah.kress@googlemail.com (H.K.); roman-klein@web.de (R.K.); 2Klinik für Unfall-, Hand- und Wiederherstellungschirurgie, Saarland University Medical Center, 66421 Homburg, Germany; tim.pohlemann@uks.eu

**Keywords:** fracture, osteoporosis, spine, pelvis

## Abstract

*Background and Objectives*: The number of geriatric patients presenting with fragility fractures of the pelvis is increasing due to ageing Western societies. There are nonoperative and several operative treatment approaches. Many of which cause prolonged hospitalisation, so patients become bedridden and lose mobility and independence. This retrospective study evaluates the postoperative outcome of a computed tomography-guided (CT-guided) minimally invasive approach of sacroiliac screw osteosynthesis. The particular focus is to demonstrate its ease of use, feasibility with the equipment of virtually every hospital and beneficial outcomes to the patients. *Materials and Methods*: 28 patients (3 men, 25 women, age 80.5 ± 6.54 years) with fragility fractures of the pelvis types II-IV presenting between August 2015 and September 2021 were retrospectively reviewed. The operation was performed using the CT of the radiology department for intraoperative visualization of screw placement. Patients only received screw osteosynthesis of the posterior pelvic ring and cannulated screws underwent cement augmentation. Outcomes measured included demographic data, fracture type, postoperative parameters and complications encountered. The quality of life (QoL) was assessed using the German version of the EQ-5D-3L. *Results*: The average operation time was 32.4 ± 9.6 min for the unilateral and 50.7 ± 17.4 for the bilateral procedure. There was no significant difference between surgeons operating (*p* = 0.12). The postoperative CT scans were used to evaluate the outcome and showed only one case of penetration (by 1 mm) of the ventral cortex, which did not require operative revision. No case of major complication was reported. Following surgery, patients were discharged after a median of 4 days (Interquartile range 3–7.5). 53.4% of the patients were discharged home or to rehabilitation. The average score on the visual analogue scale of the EQ-5D-3L evaluating the overall wellbeing was 55.6 (Interquartile range (IQR) 0–60). *Conclusions*: This study shows that the operative method is safe to use in daily practice, is readily available and causes few complications. It permits immediate postoperative mobilization and adequate pain control. Independence and good quality of life are preserved.

## 1. Introduction

Fractures of the pelvis often occur due to high energy trauma such as car accidents or falls from great heights. In the elderly population, however, they can arise after only a minor impact, such as a fall from a sitting or standing position. In some instances, a traumatic event may not even be memorable [1,2]. Due to the demographic change, these so-called insufficiency fractures are constantly increasing in number [3] and significant morbidity and mortality ensue [4].

Pelvic fractures in the elderly are also known as Fragility Fractures of the Pelvis (FFP) [5]. They arise due to low bone mineral density and are considered to be of the osteoporotic fracture entities [6]. Therefore, most patients suffering from these fractures are ageing women, as the female gender conveys a higher risk for osteoporosis [2]. Additionally, these patients more often present with comorbidities, which naturally puts them at a higher risk for complications and early death.

Patients often present with great pain and immobility. When using conventional X-ray examination, this fracture entity is often missed. Computed tomography is essential for a thorough assessment and detection of complications. Different treatment approaches are available depending on the FFP subtype (I–IV). FFP type I describes fractures of the anterior pelvic ring only. FFP type II includes non-displaced fractures of the posterior pelvic ring. Whereas FFP type III are displaced unilateral fractures of the posterior pelvic ring, FFP IV are bilateral displaced fractures of the posterior pelvic ring [7].

FFPs can be handled either nonoperative or operatively. Either approach should mainly focus on pain relief and early mobilization to reduce the rate of complications and improve the overall long-term outcome. Both treatment modes have risks and benefits. The non-operative approach, on the one hand, bears the risk of long-term immobilization, which may result in pneumonia, urinary tract infections and muscle wasting (bed rest causes a 1–1.5% loss of muscle mass every day [8]), resulting in loss of independence.

The operative approach, on the other hand, often results in earlier mobilisation but bears the risk of anaesthesia and the operation itself: hematoma, infection and impaired wound healing.

The latter favours operation modes with as minor tissue damage as possible. A minimally-invasive approach allows for a quicker discharge from the hospital and thereby has a lower risk of complications in the time shortly after surgery [9]. Several minimally-invasive approaches exist: Sacroplasty has been shown by Richards et al. to not fully restore strength or stiffness of the sacrum and cement distribution is poorly controlled [10]. Conversely, minimally-invasive screw placement across the sacroiliac joint has proven to be sufficiently stable. Some surgeons prefer introducing a transsacral bar (Bilateral screws and an additional screw on the side of the fractures) to further improve stability [11]. However, a study conducted by Gänssler et al. clearly shows that unilateral screw placement is sufficient to achieve clinical improvement [12]. We used this method and augmented the screw after placement. One cannulated screw per fractured side was embedded in PMMA (Polymethylmathacrylat) cement. This provides increased stiffness and pull-out resistance, as described by Wähnert et al. [13]. These minimally-invasive screw placements may be monitored using a C-arm (fluoroscopy) in the operation theatre. To date, this is the most widely-used mode of imaging, but as it only provides a two-dimensional view, it is more time-consuming and bears an increased risk of screw misplacement [14]. A study conducted by Gras et al. reported a 6% screw misplacement in postoperative CT scans following fluoroscopy for intraoperative imaging [14]. Alternatively, placement can be performed as a CT-navigated approach. This uses an expensive navigation system that indicates the optimal screw position. Due to its cost, this is unavailable to most surgeons [15]. A more straightforward alternative, as described here, is the CT-guided approach, where the screw-osteosynthesis is performed in a standard CT scanner for intraoperative visualisation [9]. This approach is described in the study conducted by Falzarano et al., however in comparison to our study cohort their patients were significantly younger [16].

This paper analyses the outcome of osteosynthesis of FFPs type II–IV managed with the minimally- invasive CT-guided percutaneous surgical procedure.

There are many papers published on the outcome of the different surgical procedures and several papers analyzing data on pelvic fractures in geriatric patients. However, little research has been published about this operative technique solemnly used on fragility fractures.

## 2. Materials and Methods

This study retrospectively analysed the medical records and postoperative CT imaging of patients treated in the Department of Traumatology of the Marienhaus Klinikum Hetzelstift in Neustadt an der Weinstrasse, Germany, between August 2015 and September 2021. Patients who presented with an FFP II–IV fracture and received treatment using the CT-guided percutaneous osteosynthesis of the posterior pelvic ring were included. Patients under the age of 65 years and who did not consent to participate were excluded (Figure 1).

The fractures were classified using the Fragility Fractures of the Pelvis (FFP) classification published by Rommens and Hofmann [7]. All patients with FFP II–IV were treated surgically and received the operative procedure as described below.

The operation took place in the computed tomography (CT) suite under sterile conditions in general anaesthesia. The patient was positioned sidewards. Following 3-fold skin disinfection and sterile draping, the first scan was performed. Thereby the most suitable plane for the screw (diameter 6.5 or 7.5 mm, 65 to 100 mm length, ISG screw, Marquardt Medizintechnik, Spaichingen, German) entry point and -angle was determined. A short skin incision (approximately 1 cm) was made and a bone cannula inserted in the intended screw path. Control scans were performed until adequate positioning was achieved. Then this cannula was replaced by a guiding wire, the position of which was controlled again with another scan. The screw, including a washer, was introduced manually. After removing the guided wire, augmentation with PMMA was performed and wound suturing ensued (Figure 2).

The following demographical data were collected: gender, age and comorbidities at admission. The medical records were analysed for: further information on the mechanism of injury, American Association of Anesthesiology (ASA) classification, type of screw, cement volume applied, time of operation, total hospitalisation time, postoperative hospitalisation time, postoperative pain (using the numeric rating scale (NRS)) and analgesia requirement additional to the standard pain medication SOPs, rehabilitation institution after discharge, in-hospital complications (i.e., infection, hematoma, pressure ulcers, etc.), revision surgeries and whether the patients returned to their homes after discharge. The CT images were analysed for FFP classification, screw position, distance from the cortical bone and cement leakage. Additionally, the amount of radiation (mGy*cm) was collected for every operation. The axial scan was used to analyze the distance of the screw from the neuroforamina, the sagittal plane to analyze the distance to the anterior and posterior border and the distance to the caudal and cranial border of the bone. Additionally, the cases of cement leakage and the amount of leakage were analysed using the axial, sagittal and coronal planes.

Furthermore, the postoperative quality of life (QoL) was assessed using the standardised EQ-5D-3L questionnaire [17,18]. The patients received this questionnaire at least six months after the operation. Its items enquire about five different aspects of their mobility and independence in daily living. The answers given by the patients were scored. One being independent in the activity and three being reliant on help. Patients were additionally asked to score their overall health on a visual analogue scale from 0 to 100 (zero being extremely poor health and 100 in best health).

The data was collected using Microsoft 365 Excel^®,^ (version 16.59 (22031300)). The following statistical tests were retrieved: mean, standard deviation, median, interquartile range and the one-way ANOVA (Analysis of Variance) test was employed to calculate *p*-values. A *p*-value < 0.05 was considered statistically significant.

## 3. Results

### 3.1. Demographics

Of the 28 fractures treated, 25 were of female (89.3%) and 3 of male (10.7%) gender. The mean age was 80.5 ± 6.54 years (Figure 3). The youngest patient was 67 years old, whereas the oldest was 91 years old. All patients presented with at least one comorbidity and 16 patients (57.1%) with three or more comorbidities. A comorbidity was registered as such as soon as it was mentioned in previous medical notes or the discharge letter. Previously diagnosed cardiovascular conditions were quite common conditions to be met. Hypertension was the most common, with 23 patients (82.1%) affected. The median ASA score was 3 (mean 2.7, Table 1).

Of the patients admitted, 21 presented with a unilateral fracture (75.0%). Seven patients presented with bilateral fractures. 15 incurred an FFP II (53.6%), 6 an FFP III (21.4%) and only 7 an FFP type IV fracture (25.0%). Of these 28 fractures treated, 5 followed a conscious trauma (17.9%) and 17 were considered to have insufficiency fractures for which no relevant trauma was recalled (60.1%). In 6 cases (21.4%), it could not be determined retrospectively whether the fracture occurred due to trauma or not (Table 1, Figure 4).

### 3.2. The Operative Procedure

Regardless of the FFP subtype II-IV, all patients received percutaneous osteosynthesis of the posterior pelvic ring by 5 different surgeons. None of the patients received osteosynthesis of the anterior pelvic ring. However, there was variation in the screw dimensions and whether cement was implanted. The screws varied in size from 6.5 × 75 mm to 7.5 × 100 mm. Most commonly, the 7.5 × 75 mm screw was used (39.4%). In 24 out of 33 operational procedures, PMMA (polymethylmethacrylate) cement was employed (72.7%). One patient had to be excluded as there was no documentation on the screw used during the procedure. On average, the time taken for the operation was 32.4 ± 9.6 min for one side and 50.7 ± 17.4 min when both sides were operated upon. There was no significant difference in the time required to conduct the unilateral procedure between surgeons (*p* = 0.12). The average radiation dose the patients were exposed to during the unilateral procedure was 274.0 ± 138.3 mGy*cm and for the bilateral procedure 472.0 ± 201.3 mGy*cm (Table 2).

The position of the screws was analysed using the CT scans taken at the end of the procedure. On average, the distance to the dorsal cortex was 10.1 ± 4.6 mm; there was no penetration in any case. The average distance to the ventral cortex was 4.7 ± 3.8 mm; there was a cortex penetration in one case by 1 mm. However, this did not require revision surgery. The average distance to the caudal border of the bone was 11.8 ± 5.2 mm and to the cranial border 7.1 ± 4.6 mm, with no case of penetration of the cortex. The mean distance to the cortex of the neuroforamina was 4.4 ± 3.4 mm with no case of penetration. Cement leakage could be detected in 5 of 33 operations (15.2%), but none affected nerves. Three patients had to be excluded due to missing CT-images. None of the cases required revision surgery (Table 3).

### 3.3. Postoperative Course

The median length of hospitalisation was 12 days, the shortest stay being just four and the longest 21 days (Interquartile range (IQR) 9–15.5 days). The operation was performed at a median of day six after admission due to the availability of the CT scanner. There was no significant difference between patients who presented with several comorbidities and patients with only one or no comorbidity regarding the total (*p* = 0.87) and the postoperative hospitalisation (*p* = 0.35) (Table 4 and Table 5). 18% of patients suffered from minor complications, the most frequent being urinary tract infections and bedsores. These did not influence the length of total hospitalisation. There were no in-hospital mortality, no neurological palsy or vascular lesion following surgery.

The patients’ postoperative wellbeing was assessed by evaluating the complications encountered, the pain and the analgesia requirements. The average pain stated by the patients on the NRS (numeric rating scale) was 1.32 ± 0.95 out of 10 on the first postoperative day. Only five patients (17.9%) required additional analgetics.

### 3.4. Discharge

Following surgery, patients were discharged after a median of 4 days (1–14 days, IQR 3–7.5 days). To evaluate recovery, the post-hospital destination was assessed. Ten patients (35.7%) directly went to rehabilitation, 12 patients (42.9%) were first transferred to geriatric clinics and five patients (17.9%) were discharged to the location they had been living before admission.

### 3.5. Outcome Measures

To assess the overall outcome over time, patients were asked to complete the QoL questionnaire at least six months after discharge from the hospital. Of 28 patients total, 18 returned the questionnaire. It was completed at a median of 29.5 months post-discharge. With the longest interval being 76 months and the shortest six months. When asked about general mobility, patients assessed this with a score of 1.59 ± 0.62 out of 3. Independent self-care was rated with an average score of 1.61 ± 0.71 out of 3 and activities of daily living with an average score of 1.78 ± 0.83. The pain was rated with a score of 1.82 ± 0.64 of 3 and general anxiety with 1.78 ± 0.83. The Visual Analogue Scale of overall health was completed with a mean score of 55.6 (10–95, IQR 0–60) (Table 6).

## 4. Discussion

CT-guided SI-screw osteosynthesis has been shown to be a precise, quick and safe method to be used for the treatment of fragility fractures of the pelvis of the elderly.

### 4.1. Precision

The outcome of screw position and rate of cortical bone or foramen perforation found after assessing the CT scans of our patients are also comparable to the study conducted by Reuther et al. They evaluated a similar CT-guided approach but did not limit the patient group to geriatric patients [9]. They described only a few cases where the screw lay within the cancellous bone but had no penetration. There was only one case (5.6%) of penetration of the cortical bone in our cohort. In comparison, a study conducted by Richter et al. evaluating the Computer-assisted approach using c-arm to construct a 3D-scan reported a perforation rate of 16% [20]. Notable is, that our patients’ radiation exposure was 28% less for unilateral procedures and similar for bilateral procedures conducted with CT-guidance [9].

### 4.2. Speed

We reported a shorter time for operations when compared to similar studies. Other authors report operating times up to 62 min per side using a non-augmenting technique [14]. We employ augmentation that takes approximately 5–10 min extra per screw. Furthermore, our data shows that operating time decreased by 10% when comparing the data for August 2015 to August 2020 to the data for September 2020 to September 2021 reflecting increasing experience of the individual surgeon

### 4.3. Safety

Complications encountered in our patients are similar to those described in the study by Rommens et al., the most frequent being urinary tract infection and bedsores [21]. The complications found in our patients are also comparable to the general population, as the second most common medical condition experienced by geriatric patients is urinary tract infection [22].

### 4.4. Clinical Outcomes

Additionally, the length of postoperative hospitalisation was shorter than in comparable studies looking at patients with minimally-invasive procedures, where the median hospitalisation was 12 and 17 days [14,21]. The pain reported by our patients after the operative procedure was less than in similar studies published to date [21]. Thereby the CT-guided operative method reduces the risk of complications caused by prolonged hospitalisation and supports the conservation of independent mobility.

Overall, quality of life was scored <2 for all parameters, which means that most patients can complete activities of daily living and self-care without support. A study published by Janssen et al. in 2021 evaluates the results of the visual analogue scale (VAS) of the ED-Q5 questionnaire of a broad population of Germany. For the population ≥75 years, this is 62.8 [23]. Our patients scored 55.6. This shows that their quality of life is nearly maintained and comparable to the broad population of a similar age.

## 5. Conclusions

CT-guided placement of sacroiliac screws uses an intraoperative imaging mode that is easy to learn and grants excellent control of screw positioning and cement distribution. Stabilization of the posterior pelvic ring suffices to stabilise fragility fractures providing enough strength to allow for good pain relief even if the posterior and anterior pelvic ring is fractured. This method of osteosynthesis can be carried out in nearly every institution dealing with orthopaedic trauma. Hence, it should be increasingly used to resolve the problem created by this particular fracture entity.

## Figures and Tables

**Figure 1 medicina-58-00809-f001:**
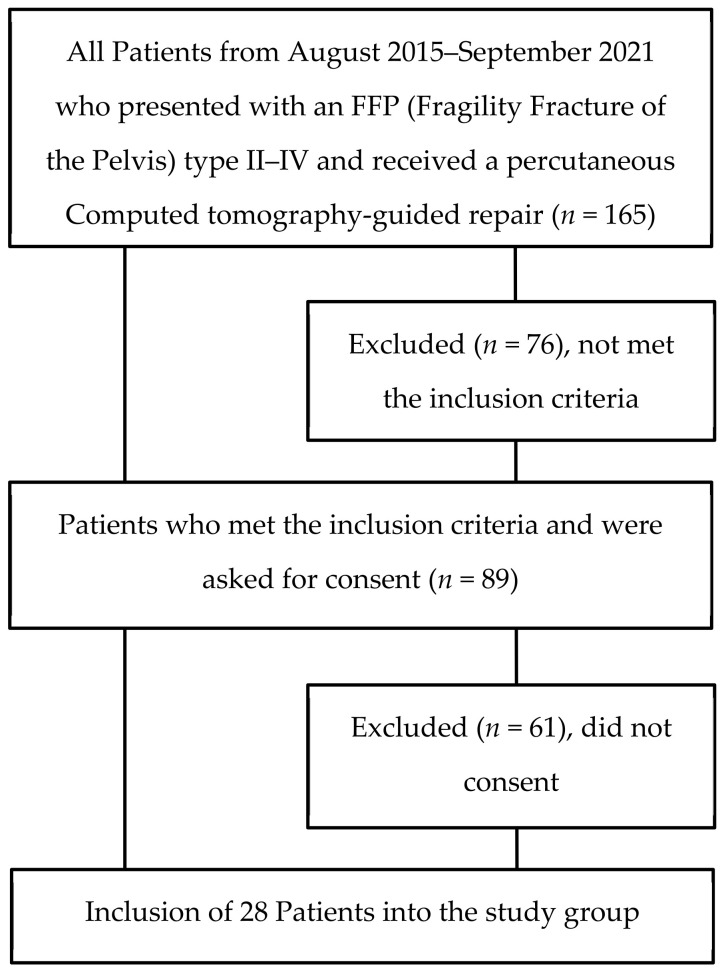
Consort Diagram.

**Figure 2 medicina-58-00809-f002:**
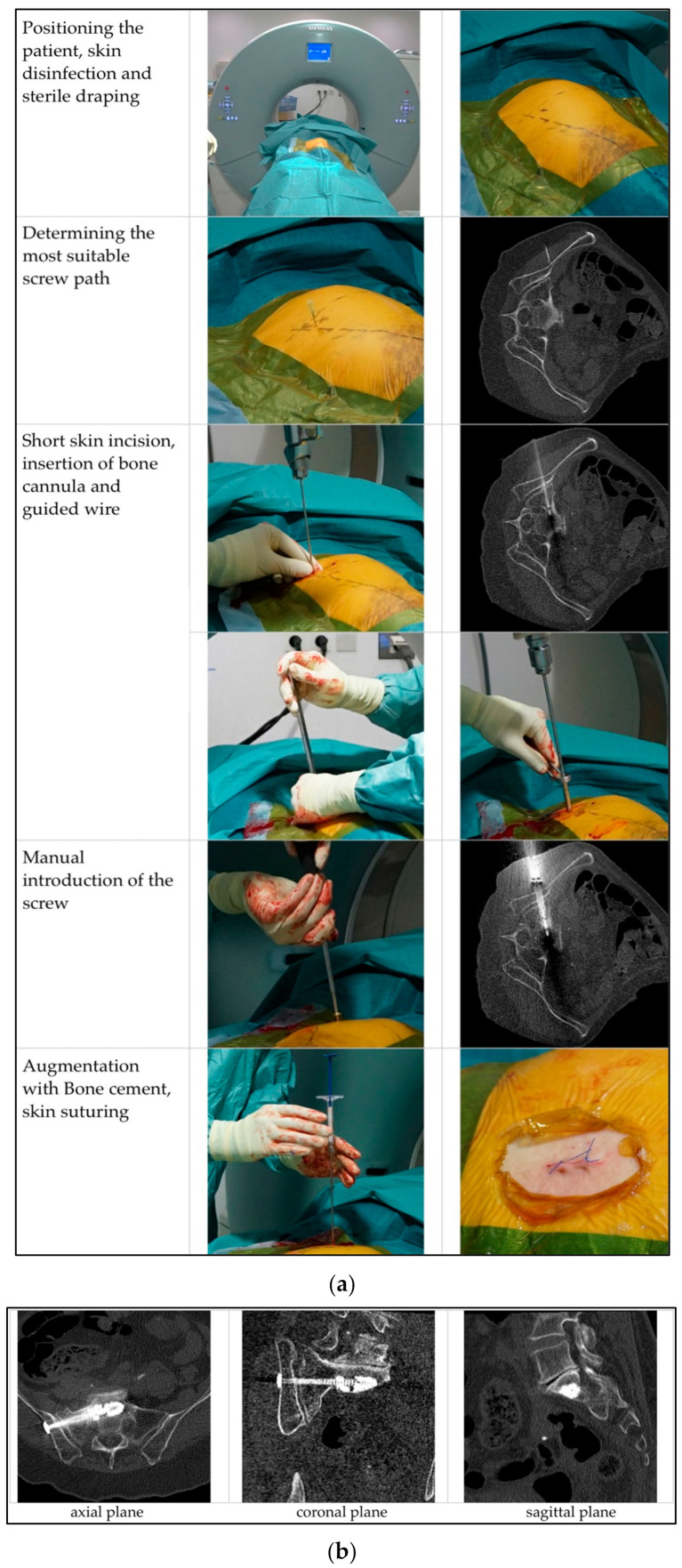
(**a**) Operative Procedure, (**b**) Post-operative CT-scans.

**Figure 3 medicina-58-00809-f003:**
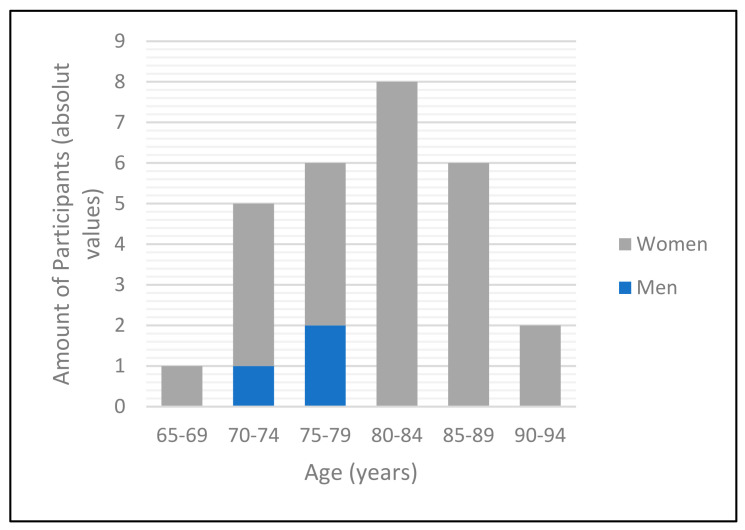
Bar diagram representing the distribution of gender and age within the study cohort.

**Figure 4 medicina-58-00809-f004:**
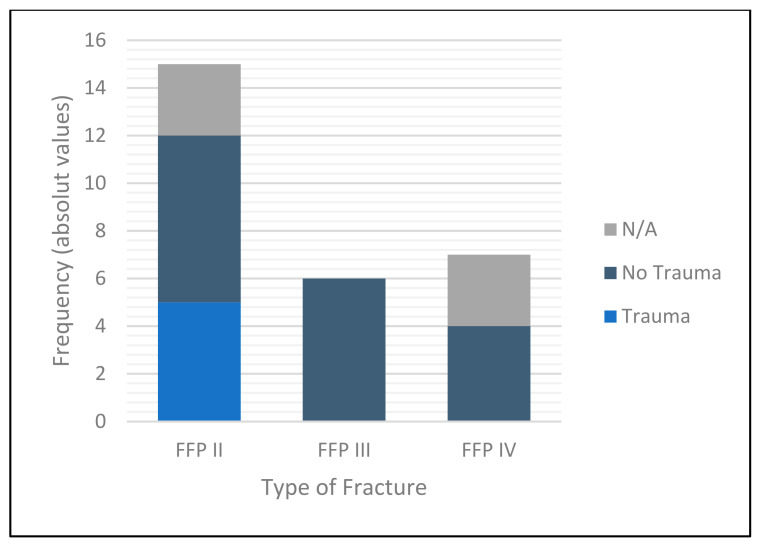
Bar diagram representing fracture classification and mechanism of injury encountered. “N/A” stands for Not applicable and “FFP” for Fragility Fractures of the Pelvis.

**Table 1 medicina-58-00809-t001:** Demographic data, fracture subtype and type of injury of our patient cohort.

Age (Years)		65–74	75–84	85–94	Total
Gender	Male	1 (3.57%)	2 (7.14%)	0 (0.00%)	3 (10.71%)
	Female	5 (17.86%)	12 (42.86%)	8 (28.57%)	25 (89.29%)
ASA ^1^ score	1	0	0	0	0 (0.00%)
	2	1	6	1	8 (28.57%)
	3	5	8	7	20 (71.42%)
	4	0	0	0	0 (0.00%)
	5	0	0	0	0 (0.00%)
Comorbidities	Average number of comorbidities	3.67 ± 1.51	3.07 ± 1.54	3.63 ± 1.51	3.57 ± 1.50
	art. hypertension	3	13	7	23 (82.14%)
	chronic pain syndrome	3	2	1	6 (21.42%)
	coronary heart disease	1	2	6	9 (32.14%)
	diabetes mellitus	2	1	1	4 (14.29%)
	obesity	2	4	3	9 (32.14%)
	atrial fibrillation	1	1	2	4 (14.29%)
	Others	5	7	6	18 (64.29%)
Fracture Type	FFP IIa ^2^	0	0	0	0 (0.00%)
	FFP IIb ^3^	4	8	3	15 (53.47%)
	FFP IIc ^4^	0	0	0	0 (0.00%)
	FFP IIIa ^5^	0	1	1	2 (7.14%)
	FFP IIIb ^6^	0	1	1	2 (7.14%)
	FFP IIIc ^7^	0	0	2	2 (7.14%)
	FFP Iva ^8^	0	0	0	0 (0.00%)
	FFP IVb ^9^	0	1	1	2 (7.14%)
	FFP IVc ^10^	2	3	0	5 (17.86%)
Mechanism of Injury	Trauma	2	3	0	5 (17.85%)
	No Trauma	3	6	8	17 (60.71%)
	Not classified	1	5	0	6 (21.43%)

^1^ American Society of Anethesiologists. ^2^ Fragility Fractures of the Pelvis type IIa is an injury only to the dorsal posterior pelvic ring, which is non-displaced. ^3^ Fragility Fractures of the Pelvis type IIb is a sacral crush fracture with anterior disruption. ^4^ Fragility Fractures of the Pelvis type IIc is a non-displaced sacral, sacroiliac or iliac fracture with anterior disruption. ^5^ Fragility Fracture of the Pelvis type IIIa is a displaced unilateral ilium fracture with anterior disruption. ^6^ Fragility Fracture of the Pelvis type IIIb is a displaced unilateral sacroiliac disruption with anterior disruption, ^7^ Fragility Fractures of the Pelvis type IIIc is a displaced unilateral sacral fracture together with an anterior disruption. ^8^ Fragility Fractures of the Pelvis type IVa are bilateral iliac fractures or bilateral sacroiliac disruptions together with an anterior disruption. ^9^ Fragility Fractures of the Pelvis type IVb is a spinopelvic dissociation with anterior disruption. ^10^ Fragility Fractures of the Pelvis type IVc is a combination of different posterior instabilities together with anterior disruption [19].

**Table 2 medicina-58-00809-t002:** Outcomes of unilateral and bilateral surgery (“surgeon 2” did not perform any unilateral procedure, hence was excluded from this table).

Surgeon	Number of Procedures	Average Time (min)	Radiation Exposure (mGy*cm)
		Unilateral procedure	
1	5	40	558.14
3	14	30	264.72
4	2	30	236.60
5	0	N/A ^1^	N/A
		Mean 32.38 ± 9.57 (*p* = 0.12)	Mean 265.17 ± 142.68 (*p* = 0.61)
Bilateral procedures			
1	2	40	379.00
2	1	55	780.00
3	2	45	267.54
4	1	75	287.00
5	1	55	N/A
		Mean 50.71 ± 17.42	Mean 393.35 ± 201.30

^1^ Not Applicable.

**Table 3 medicina-58-00809-t003:** Data gained from the postoperative CT-images. Measurement of distance from the cortex.

Distance (mm) to:	Posterior Cortex	Anterior Cortex	Caudal Cortex	Cranial Cortex	Neuroforamina
Average values (mm)	10.14 ± 4.54	4.69 ± 3.77	11.75 ± 4.61	7.09 ± 4.61	4.33 ± 3.44

**Table 4 medicina-58-00809-t004:** Total and postoperative hospitalisation in the different age groups.

Age		65–74	75–84	85–94	Total
Total hospitalization (days)	Mean	10.17	13.36	11.25	12.07
	Standard deviation	±6.52	±4.24	±3.30	±4.59
	Median	8.00	13.00	12.00	12.00
	IQR ^1^	5–16	11–16	9–14	9–15.5
					*p* = 0.32
Postoperative hospitalization (days)	Mean	5.83	6.71	3.00	5.21
	Standard deviation	±2.73	±3.77	±2.00	±3.46
	Median	3.50	9.50	3.00	4.00
	IQR	3–7	4–10	1–4.5	3–7.5
					*p* = 0.04

^1^ Interquartile range.

**Table 5 medicina-58-00809-t005:** Total and postoperative hospitalisation of patients who received unilateral compared to a bilateral surgery.

Age		Unilateral Procedure	Bilateral Procedure	Total
Total hospitalization (days)	Mean	12.43	11.00	12.07
	Standard deviation	±4.30	±5.60	±4.59
	Median	12.00	9.00	12.00
	IQR	9.5–15.5	7–16	9–15.5
				*p* = 0.50
Postoperative hospitalization (days)	Mean	10.17	13.36	5.21
	Standard deviation	±6.52	±4.24	±3.46
	Median	8.00	13.00	4.00
	IQR	5–16	11–16	3–7.5
				*p* = 0.50

**Table 6 medicina-58-00809-t006:** Replies of the ED-Q5 questionnaire categorised by time interval post-surgical intervention.

Time Interval Post-Surgery (Months)	6	7–12	13–23	24–35	36–47	48–59	>60
Quantity (absolute values)	1	3	6	1	1	2	3
Mobility	1.00	1.67	1.50	2.00	3.00	1.50	1.67
Self-Care	1.00	1.67	1.17	2.00	3.00	2.00	1.67
Usual activities	1.00	1.67	1.17	3.00	3.00	2.50	2.00
Pain/Discomfort	2.00	1.67	1.67	2.00	2.00	2.00	2.00
Anxiety Depression	1.00	2.00	1.67	1.00	3.00	2.00	1.67
Visual Analogue Scale	85.00	75.00	66.67	50.00	10.00	35.00	25.00

## Data Availability

Not applicable.

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
