# Peer review of "Sacroiliac Screw Placement with Ease: CT-Guided Pelvic Fracture Osteosynthesis in the Elderly"

_medicina, 2022, doi:10.3390/medicina58060809_

Round 1

Reviewer 1 Report

Good paper. Just a few things that are all word changes (as is so common with papers where English is not the first language).

Abstract Line 16 and Introduction Line 63 - The word "conservative" is used when the correct term is "nonoperative". Please change.

Introduction - Line 79 - the word "additionally" should be 'additional". Please change.

Conclusion - The last sentence makes no sense. Please rewrite to make sense. (just English translation makes it difficult to interpret as grammar and word choice can be improved).

Author Response

Thank you very much for your review, I included the mentioned points below:

  • I changed the words and spelling as proposed
  • I rewrote the last sentence of the conclusion: „This method of osteosynthesis can be carried out in nearly every institution dealing with orthopaedic trauma. Hence, it should be increasingly used to resolve the problem created by this particular fracture entity.“

Reviewer 2 Report

This is very interesting topic, as these fractures are quite often missed.

However some points to review, text is attached. The most important thing is need to clear up, that in CONSORT diagram (Fig.1) are presented 27 patients, who gave consent, but in Table 1 is concentrated analysis for 28 fractures, where by my best understanding fracture is equaled with patient (as bilateral fractures are gr IV).    

Author Response

Thank you very much for your detailed review. I included the mentioned points below:

  • I Edited the grammar and spelling mistakes as highlighted in the document
  • Consort Diagramm – one patient presented on two occasions with fractures of either side of the pelvis. Initially we wanted to include him as only one patient but use the data as seperate sets, as both were unilateral fractures. However, I already noticed in the process that this may be too confusion. Therefore, I now edited the consort diagram. I decided to enroll this patient as two patients due to the two cases.
  • Excluded Surgeon 2 from Table 2.1 as proposed in the document
  • Renamed the paragraph as „outcomes measured“.
  • Included a table as proposed, demonstrating the data gained from the QoL questionnaire. I decided to format this in intervals of six months

Reviewer 3 Report

The paper provides information, how the CT-guided stabilization of the sacro-iliac joint should be. The paper approximates minimally invasive, CT-guided technique of sacroiliac screw fixation. I think that it is interesting for clinicians.It describes the surgical technique developed supposable (the most probable) by others. Nevertheless, it approximates the prolem to the wide audience and thus is interesting.

Nevertheless, I also gave you an information that the technique itself is not new, it was described previously and it originates on the idea of technique' investigators working for Stryker. Due to lately mentioned backgrounds (Falzarano G, Rollo G, Bisaccia M, et al. Percutaneous screws CT guided to fix sacroiliac joint in tile C pelvic injury. Outcomes at 5 years of follow-up. SICOT J. 2018;4:52. doi:10.1051/sicotj/2018047.) the idea presented in it is "generic"

The paper is well written - I have no doubt about its idea, the method of data presentation and obtained results.

Authors should specify the type of screw system used in their study

Author Response

Thank you very much for your review. I included the mentioned paper published by Falzarano et al. in the introduction. 

Additionally I added the producer and make of the screws used.